# Implementation of dynamic snow density within GlobSnow snow water equivalent retrieval methodology

Pinja Venäläinen[1], Kari Luojus[1], Juha Lemmetyinen[1], Jouni Pulliainen[1], Mikko Moisander[1], Matias Takala[1]

[1]Finnish Meteorological Institute, PO Box 503, FIN-00101 Helsinki, Finland.

*Correspondence to*: Pinja Venäläinen (pinja.venalainen@fmi.fi)

**Abstract.** Snow water equivalent (SWE) is an important variable in describing global seasonal snow cover. Traditionally, SWE has been measured manually at snow transects or using observations from weather stations. However, these measurements have a poor spatial coverage, and a good alternative to in-situ measurements is to use spaceborne passive

microwave observations, which can provide global coverage at daily timescales. The reliability and accuracy of SWE estimates made using spaceborne microwave radiometer data can be improved by assimilating radiometer observations with weather station snow depth observations as done in the GlobSnow SWE retrieval methodology. However, one possible source of uncertainty in the GlobSnow SWE retrieval approach is the constant snow density used in modelling emission of snow. In this paper, three versions of spatially and temporally varying snow density fields were implemented using snow transect data from

Eurasia and Canada and automated snow observations from USA. Snow density fields were used to post-process the baseline GlobSnow v.3.0 SWE product. Decadal snow density information, i.e. fields where snow density for each day of the year was taken as the mean calculated for the corresponding day over ten years, was found to produces the best results. Overall, post-processing GlobSnow SWE retrieval with dynamic snow density information improved overestimation of small SWE values and underestimation of large SWE values, though underestimation of SWE values larger than 175 mm was still significant.

## 20 1 Introduction

Snow water equivalent (SWE) is an important property of the seasonal snow cover and estimates of SWE are required in many hydrological and climatological applications, including climate model evaluation (Mudryk et al., 2018) and forecasting freshwater availability. Maximum SWE before the start of spring snow melt is one of the most important snow characteristics for run-off and river discharge forecasts (Barnett et al., 2005; Barry, 2002).


Snow depth or SWE can be estimated by interpolating surface snow depth (Dyer and Mote, 2006) or snowfall measurements (Broxton et al., 2016). However, the limited spatial and temporal coverages of the ground-based measurements, especially in northern and alpine regions, limit the quality of estimates (Broxton et al., 2016; Mortimer et al., 2020). An alternative approach for estimating SWE is to use satellite measurements as they can provide global spatial coverage and good temporal resolution.

Spaceborne passive microwave radiometer (for example, Chang and Foster, 1987; Kelly et al., 2003; Pulliainen, 2006) or active radar (for example, Lievens et al., 2019; Rott et al., 2010) observations can be used for retrieving SWE information. Passive microwave observations are commonly used as these provide frequent repeat coverage and the influence of atmospheric conditions on the observation is limited. Furthermore, passive microwave radiometer data are also available from 1978 onwards, which allows the analysis of long time series. Many passive microwave radiometer-based approaches for estimating SWE are adopted from an algorithm proposed by Chang and Foster (1987) for estimating snow depth from horizontally polarized Scanning Multichannel Microwave Radiometer (SMMR) measurements. The algorithm is based on the difference in measured brightness temperatures at a frequency insensitive to dry snow, around 19 GHz, and at a frequency sensitive to dry snow, around 37 GHz. The uncertainty of SWE retrievals based on the radiometer measurements alone can be quite high (Mudryk et al., 2015). Retrieval algorithms that use only radiometer data tend to underestimate SWE in deep snow conditions (Derksen et al., 2005) and the performance of these algorithms is even more limited in wet snow conditions (Armstrong and Brodzik, 2001).

To overcome the problems connected to stand-alone passive microwave SWE retrievals, ground-based observations, and satellite radiometer data can be combined as done in the assimilation approach for SWE retrieval introduced by Pulliainen (2006) and complemented by Takala et al. (2011). This assimilation-based approach was used as the baseline method for the Global Snow Monitoring for Climate Research (GlobSnow) initiative of the European Space Agency (ESA). The GlobSnow method has been shown to produce good results when compared to typical stand-alone radiometer algorithms (Mortimer et al., 2020). The GlobSnow version 3.0 (GSv3.0) climate data record with spatial bias correction was used for accurate reconstruction of the northern hemisphere snow mass and its trends for period of the 1979-2018 (Pulliainen et al., 2020). Improving the GlobSnow SWE retrieval methodology will help to further enhance our understanding of the northern hemisphere snow conditions and its changes.

The GlobSnow SWE retrieval utilizes a fixed density of 240 kg m$^{-3}$ throughout the retrieval regardless of snow depth, location, or time of the year (Takala et al., 2011), which is a known source of uncertainty in the retrieval. The density of snow changes with time and place, and it is greatly affected by surrounding weather conditions. For example, wind breaks down snow crystals, both on the ground and falling from the sky, which allows snow crystals to pack together tightly and increases the density of snow (Jordan et al., 1999). The age of the snow cover also affects its density as the snow on the ground is constantly undergoing metamorphism (Maurice and Harold, 1981).

One approach considered for GlobSnow SWE retrieval methodology for estimating temporally and spatially varying snow densities was to use a statistical snow density model presented by Sturm et al. (2010) which predicts density of snow as a function of the snow depth, day-of-the-year, and snow class (Luojus et al., 2013b). However, applying densities obtained using

this approach did not improve retrieval skill notably (Luojus et al., 2013a). A different approach for obtaining varying snow density information is to use available snow density data to create snow density fields by applying temporal and spatial interpolation.

In this study, dynamic snow density information obtained from ground measurements using interpolation is used to post-process the GSv3.0 SWE climate data record. Three different versions of the snow densities are implemented for Eurasia for the years 2000 to 2009. Additionally, one version of the dynamic snow densities is also implemented for the whole northern hemisphere for the whole period of GSv3.0 SWE data record, 1979-2018.

## 2 Data and methods

### 2.1 Snow density and SWE data

#### 2.1.1 Eurasia

In-situ snow density and SWE measurements were used to obtain dynamic snow density fields and validate the results of SWE retrievals. SWE and density datasets for Eurasia were obtained from Russia (Bulygina et al. 2011) and Finland (Haberkorn, 2019). These datasets contain snow transect data. Snow transects consist of manual gravimetric snow measurements made at multiple locations along a pre-defined transect several hundreds of meters to several kilometres in length which are averaged together to obtain a single representative (in regard to the spatial resolution of utilized passive microwave radiometers) SWE value for a given transect on a given date.

Russia data, from a substantial network of snow transects, has been made available via the All-Russia Research Institute of Hydrometeorological Information- World Data Center (RIHMI-WDC) website. This Russian snow survey dataset contains data from routine snow surveys operated at 515 meteorological station locations and data is available from 1966 to 2019 (Bulygina et al. 2011); data from 1979 to 2018 is used in this study. Routine snow surveys are run through the cold season every ten days or every five days during the intense snowmelt season. The Finnish Environment Institute (SYKE) has a network of about 160 snow survey courses that have been operated from the beginning of the 20th century (Haberkorn, 2019). These 2 to 4 km long snow survey courses that go through different landscapes are visited monthly, and 80 snow depth measurements are made along the snow course through varying landscape about 50 meters apart, eight snow density and SWE measurements are made along each snow course. An aggregate of the SWE measurements is applied to describe the SWE conditions for the snow course for the given sampling date.

The Russian snow transect dataset was divided into two parts. The division of data was done by finding the nearest neighbours and separating them into different datasets. The first part of the data was used for implementing the dynamic snow density maps and the second part, together with snow transect data from Finland, was used for validating snow density and SWE

95    results. The implementation dataset contains 257 locations, and the validation data is formed from data from 625 locations. Implementation and validation snow transect locations are shown in Fig. 1. Figure 2 shows histograms of the implementation and validation snow density values.

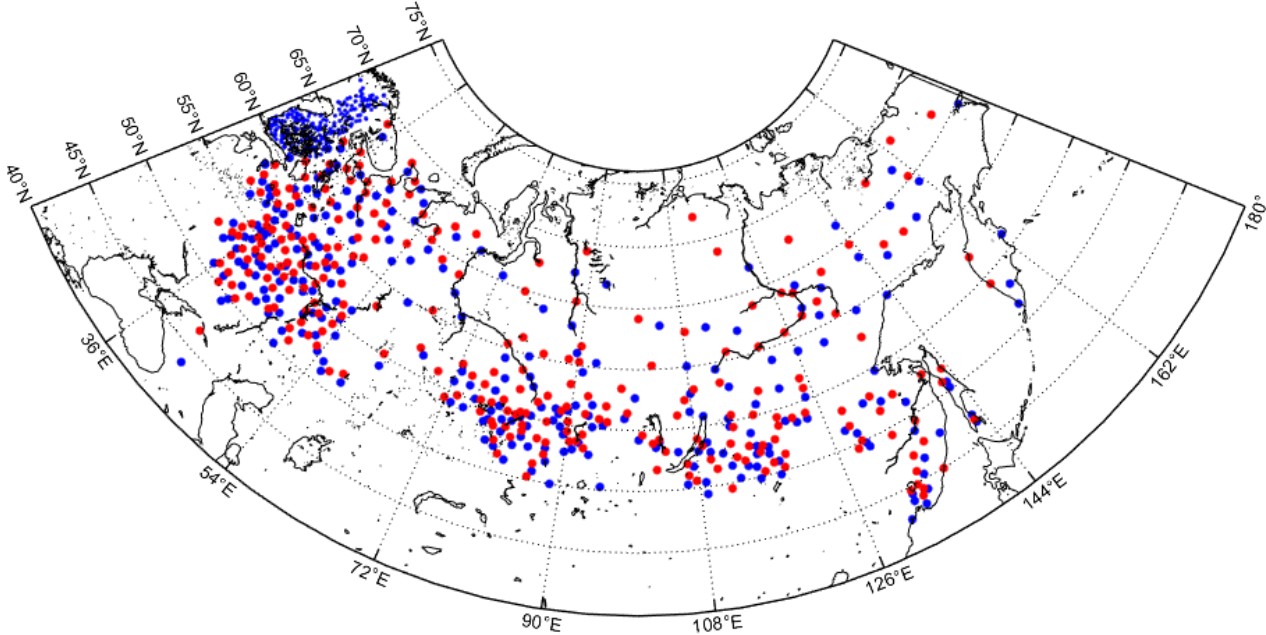

**Figure 1**. Locations of Eurasian snow courses divided into two sets: implementation (red) and validation (blue).

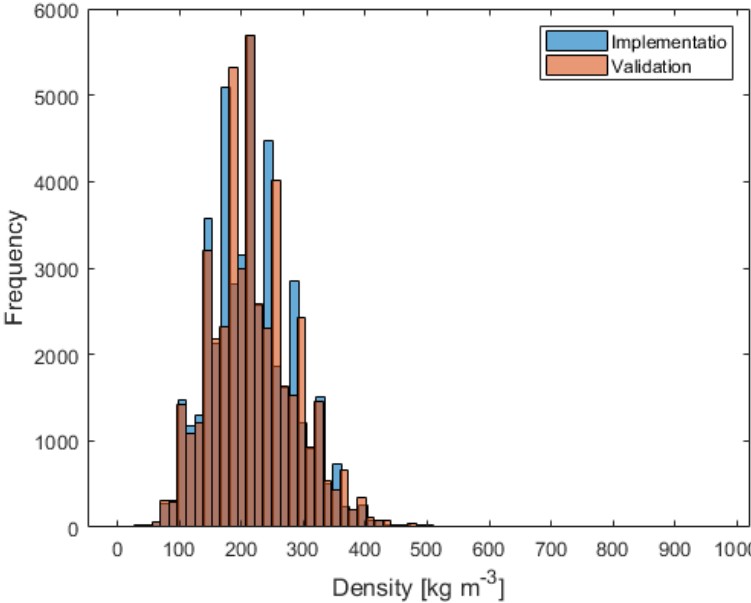

100

**Figure 2**. Histogram of the implementation and validation densities for Eurasia for 2000-2009.

We divided the Eurasian snow density implementation dataset into three distinct ways to create three different versions of dynamic snow density maps. For the first version, called the multi-decadal version, all data between 1979 and 2018 were used. The second version, called the decadal version, uses data from 2000 to 2009. The third version, called the annual version, uses data from 2000 to 2009 and produces daily density maps for each year using only data from the year under investigation.

A day-of-the-winter (DOW) value was added to each snow density measurement. DOW values are a modification of the day-of-the-year (DOY) values. DOW values start from September 1 and then continue to grow from there until the last day of August. This means that January 1 has a DOW value of 123, and August 30 has a value of 365 or 366. They are used because in the northern hemisphere winter season spans from September until June over the new year.

The Eurasian snow survey data was filtered to remove all negative density observations and all observations larger than 1000 kg m$^{-3}$. These measurements are most likely erroneous as snow densities typically range between 50 and 550 kg m$^{-3}$ (Fierz et al., 2009). After filtering, average snow density values were calculated for each DOW that had at least one measurement.

Most density measurements have been done systematically on the same DOW from year to year, with few exceptions. This difference in measurement days may cause average densities to fluctuate from one day to another as some density values are not averages but measurements from one specific year. To avoid these fluctuations in multi-decadal and decadal versions, if two consecutive DOWs had measurements, the average density is calculated using data from both days, and this density was assigned the DOW value of the first day. Outlier data points were removed from multi-decadal and decadal datasets after averaged densities were calculated. Outliers were determined to be data points that differ from two previous and two following points by more than 50 kg m$^{-3}$. Figure 3 shows how average densities calculated for each DOW and non-consecutive DOWs differ for one snow transect location for the multi-decadal dataset. The depicted snow transect is located in western Russia. Figure 3 shows also the 40-year average SWE for each DOW for the same station.

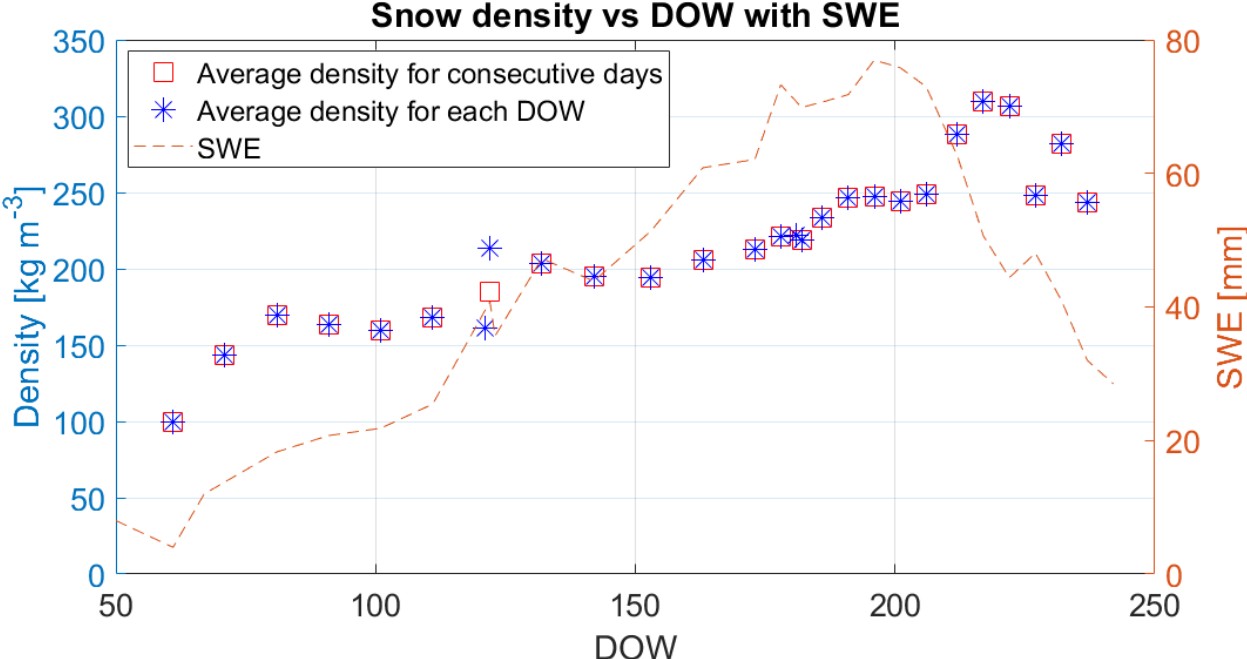

**Figure 3.** Average snow density and SWE versus DOW for snow transect in western Russia with latitude 59.4° N and longitude 33.1° E using multi-decadal data. Blue asterisks show average density calculated for each DOW separately. Red square markers show average densities calculated using data for two consecutive days if available. The red dashed line shows averaged SWE for the same location.

### 2.1.2 North America

The North American dataset consists of data from Canada and the USA. The Canadian snow course network is sampled twice per month (around the first and 15th) during the snow season and the dataset extends from 1981-2016 (Brown et al. 2019). The Canadian dataset was complemented with snow observations from 443 SNOTEL stations located in Alaska and the north-western United States (Serreze et al., 1999). Data from southern states is not included as most of the snow in these areas is in mountains which are excluded from the retrieval. The SNOTEL dataset differs from the Canadian and Eurasian datasets, as it consists of automated daily measurements instead of manual snow transect measurements. SNOTEL stations collect data on snowpack SWE, snow depth, precipitation, and air temperature. SWE is measured by a snow pillow filled with an antifreeze solution. Hourly data are available from the snow pillows, but daily measurements were used as they are more robust as hourly data is easily affected by wind and sensor issues.

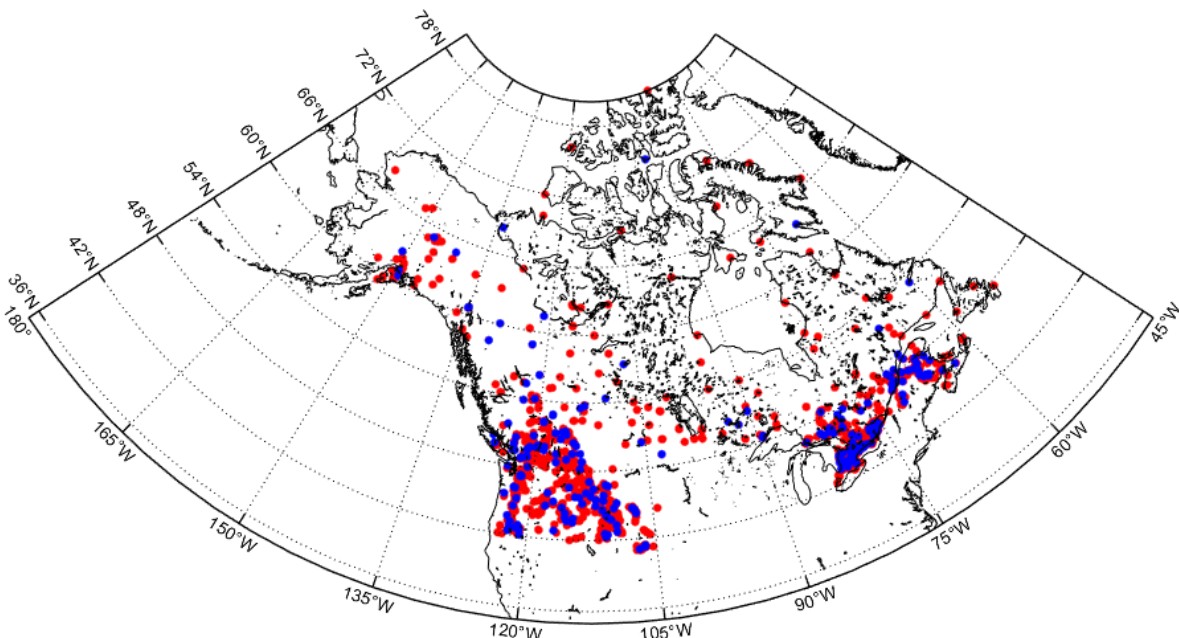

**Figure 4.** Locations of north American snow measurements divided into two sets: implementation (red) and validation (blue).

A small part of North American data was separated to be used for validation of results. The implementation set contains 1455 locations and the validation set is made from 242 locations. The locations that form the validation and implementation datasets are shown in Fig. 4.

**2.2 Baseline SWE retrieval**

The European Space Agency (ESA) GlobSnow and succeeding projects have produced a family of daily satellite-based SWE
climate data records spanning over 40 years. The most recent GSv3.0 data record is based on methodology introduced in Pulliainen (2006) and Takala et al. (2011) and the latest version is presented in detail in Luojus et al. (2021). The retrieval algorithm combines satellite-based passive microwave measurements with ground-based synoptic weather station snow depth observations by Bayesian non-linear iterative assimilation.

The GlobSnow approach uses two vertically polarized brightness temperature observations at 19 and 37 GHz and a scene brightness temperature model (the HUT snow emission model (Pulliainen et al., 1999)). First, an effective snow grain size is estimated for grid cells that coincide with weather station snow depth observations. The snow grain sizes are used to construct a kriging interpolated background map of the effective grain size, including an estimate of the effective grain size error. This spatially continuous map of grain size is then used as an input for HUT model inversion to provide an estimate of SWE. The
daily weather station snow depth measurements are also used to form a continuous background field of snow depth

independently from passive microwave measurements. The interpolated snow depth field is fused with space-borne brightness temperature observations, using the scene brightness temperature model in a Bayesian approach that weights all information sources with their estimated variances, to provide the final SWE estimates. A constant value of snow density is used (240 kg m$^{-3}$).


The retrieval method does not produce SWE estimates for mountainous areas, glaciers, or Greenland. The data record is based on data from the SMMR aboard NIMBUS-7, and SSM/I and SSMIS sensors onboard DMSP 5D F-series satellites, and synoptic weather station snow depth data from the northern hemisphere.

## 2.3 Creation of snow density fields

Two main steps for generating snow density fields are temporal and spatial interpolation. Snow density measurements are made usually every 10 or 15 days and thus, there are many days without snow density observations. Simple linear interpolation was used to obtain estimates of snow density values for the days lacking observations.

The length of the snow season depends on the year and place but to get similar series for each station, and for each version,
the average of three first existing densities was added to DOW 30 if a station did not have measurements from DOW 30 or before this day. Similarly, if the station did not have any measurement after DOW 280, the average of the last three densities was calculated and added as the density for DOW 280. After this procedure, all stations have density values from DOW 30 to DOW 280, and interpolation could be performed for this period. Figure 5 shows the results of interpolation for all three versions for one snow transect location. The interpolation of the multi-decadal dataset produces smoothest results and yearly data shows
the most fluctuation.

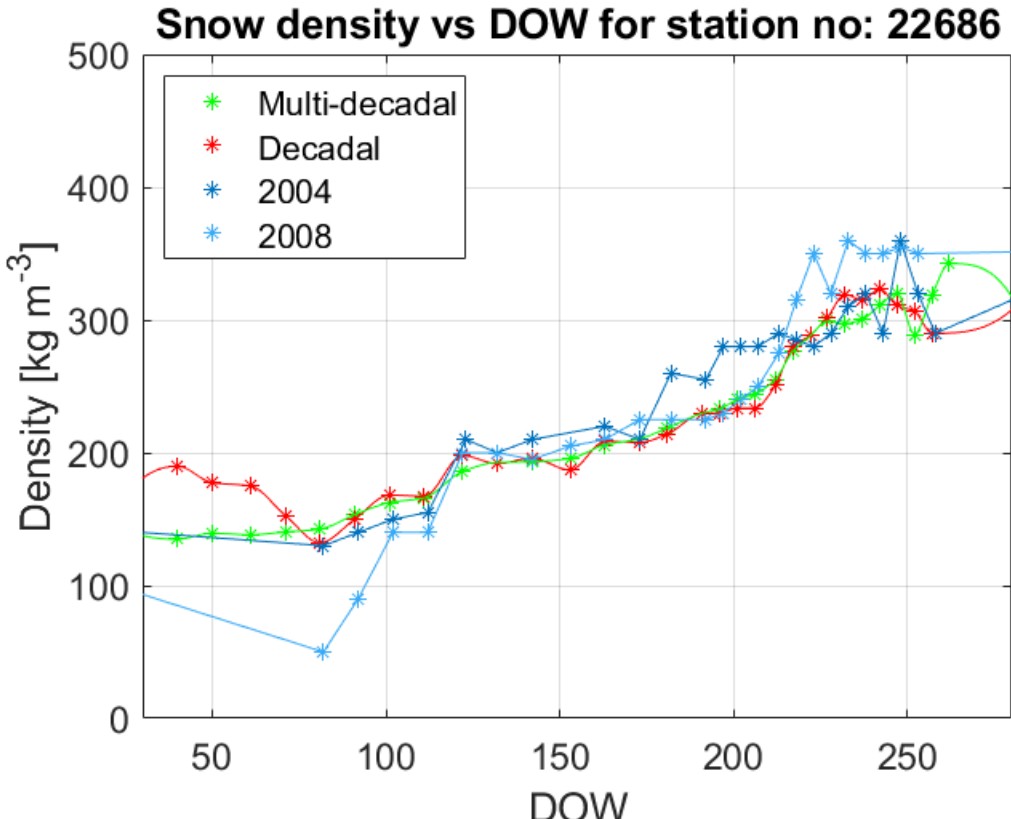

**Figure 5**. Temporally interpolated densities for snow transects in north central Russia for multi-decadal and decadal versions. Figure shows also results of interpolation for two years of annual interpolation, 2004 and 2008.

We used ordinary kriging interpolation for interpolating snow density values for areas without observations. Kriging interpolation is a spatial interpolation method that predicts values for the location with no measurements based on the spatial autocorrelation of measured values (Goovaerts, 1997). This means that two closely located points are more likely to have similar values than two points further afield. The main advantages of kriging interpolation compared to nongeostatistical interpolation methods are that kriging interpolation provides variance of predicted values and that the spatial smoothing is

defined through the variogram. The model of spatial variability can be expressed as (Høst, 1999):

$$Z(s) = \mu(s) + \epsilon(s), \tag{1}$$

where $Z(s)$ denotes the predicted value at some location, $\mu(s)$ is the deterministic function describing the trend component of $Z(s)$, and $\epsilon(s)$ is the stochastic locally varying but spatially dependent residuals.

The spatial autocorrelation is modelled with a semivariogram (Goovaerts, 1997):

$$2\gamma(h) = Var\big(Z(s_i) - Z(s_i + d)\big), \tag{2}$$

with assumption of intrinsic stationarity (variance on the right-hand side is only dependent on the vector difference d) and assumption that the process is isotropic (the autocorrelation is only dependent on the distance between observations). The empirical semivariogram can be estimated from the observations as follows (O'Sullivan and Unwin, 2010):

$$\hat{\gamma}(\tilde{h}_j) = \frac{1}{2N_d} \sum_{i=1}^{N_d} E\left(Z(s_i) - Z(s_i + d)\right), \forall h \in \tilde{d}_j, \tag{3}$$

where $Z(s_i)$ and $Z(s_i + d)$ are sampled data pairs at distance $d$ and $\tilde{d}_j$ is the distance between observations. In this study, exponential function is used for the variogram:

$$\gamma(d) = c_1 * \exp(d * c_2) + c_0. \tag{4}$$

Snow density values were predicted for each pixel in 25 km Equal Area Scalable Grid (EASE-Grid version1, to match GSv3 processor grid) between latitudes from 42° N to 80° N, and for longitudes from 20° E to 180° E. The snow density values were estimated using coordinates for centres of each pixel. Density maps were made for all three versions of snow densities for each day starting from DOW 30 and ending at DOW 280. Pixels that are not on land are assigned a value of -1, and land areas outside the area of interpolation are designated a constant density of 240 kg m$^{-3}$ (effectively retaining the original retrieval methodology for those regions where snow density could not be reliably established).

## 2.4 Usage of snow density information and validation

The derived snow density information is used to post-process the baseline GSv3.0 SWE retrieval. Post processing of SWE retrieval means that SWE values obtained are scaled with the ratio of dynamic and constant snow density:

$$SWE_{new} = SWE * \frac{\rho_{dynamic}}{\rho_{constant}}, \tag{5}$$

where $\rho_{constant}$ has values of 240 kg m$^{-3}$. Scaling is performed for each pixel within the area for which dynamic densities are available, for the regions outside dynamic snow density information, the constant density consideration is retained.

The obtained snow densities and post-processed SWE datasets were validated using independent validation data. Validation locations were separated from the data used for generating snow density fields to ensure independent cross-validation. Root-mean-squared (RMS) error, bias, correlation coefficients and mean absolute error (MAE) are the four statistical measures used for assessing the performance.

# 3 Results

## 3.1 Eurasia 2000-2009

### 3.1.1 Snow density

**Table 1.** Summary of calculated validation parameters for three snow density sets for years 2000-2009.

|  | Bias [kg m$^{-3}$] | RMSE [kg m$^{-3}$] | MAE [kg m$^{-3}$] | Correlation coefficient |
|---|---|---|---|---|
| Multi-decadal | 2.7 | 48.4 | 35.8 | 0.71 |
| Decadal | 0.9 | 48.8 | 35.8 | 0.71 |
| Annual | -2.2 | 45.0 | 32.3 | 0.76 |

Annual, decadal, and multi-decadal versions of snow density fields were produced for Eurasia for the years 2000-2009. These three versions of dynamic snow densities were compared to validation snow density data from Eurasia over the same 10-year period. A summary of the validation is shown in Table 1. Figure 6 shows the comparison of estimated and observed densities at validation sites for multi-decadal, decadal, and annual snow densities. As Fig,2 shows, most snow density values range between 150 kg m$^{-3}$ and 350 kg m$^{-3}$, which can explain the departure from 1:1 fit for small and large snow density values seen in Fig, 6.

Multi-decadal and decadal version of snow densities exhibit similar behaviour with the multi-decadal version having slightly smaller RMSE and larger correlation coefficient but larger bias than the decadal version. MAE is equal for these two versions. The annual version differs from the other two versions more and it estimates snow densities below 200 kg m$^{-3}$ and above 300 kg m$^{-3}$ better than the other versions. The multi-decadal and decadal versions of snow densities are produced from data that are averaged from a large number of measurements. This averaging means that the highest and lowest measurements have a diminished effect on estimated snow densities. Annual densities have a wider range of densities present in the data used for deriving these dynamic densities compared to the range of densities used for deriving the other two sets of the density maps. Annual densities also have a negative bias, while the other two versions have positive biases.

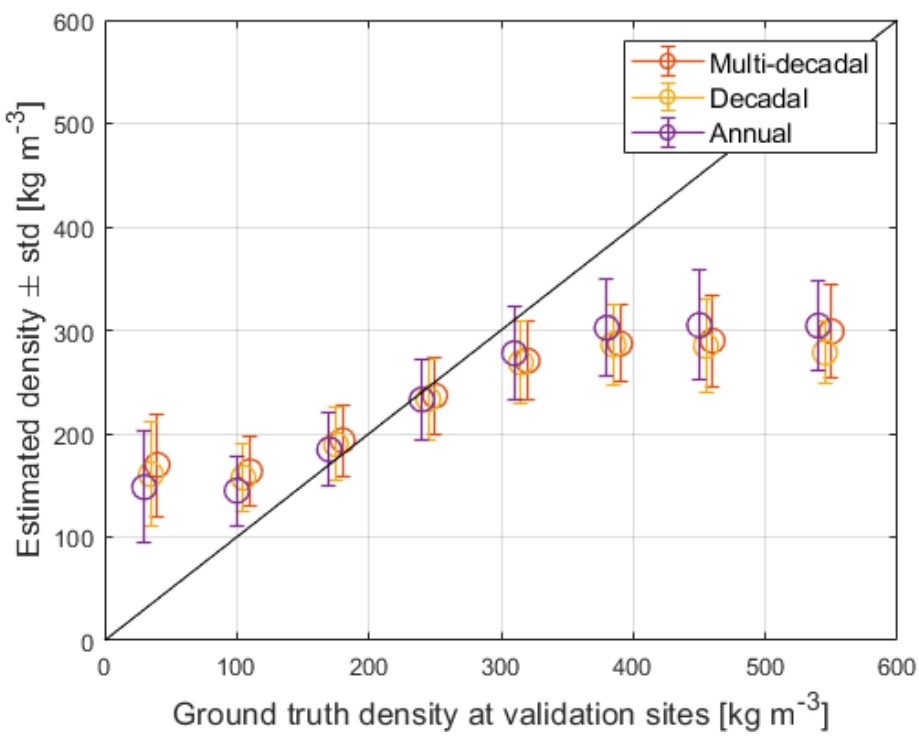

**Figure 6.** Comparison of multi-decadal, decal and annual density estimates. As can be observed, annual densities perform best for small and large density values.

### 3.1.2 Post-processed SWE

The three different density sets were used to post-process the baseline GSv3.0 SWE data between 2000 and 2009. Obtained SWE values were compared to validation SWE measurements of the Eurasian dataset. Two validation were performed, the
first validation took into account SWE values up to 500 mm and the second validation considered SWE values only up to 150 mm, as the bulk of the observations are below this value. Table 2 summarizes the results of the SWE validation (for different snow density realizations). Figure 7 shows mean errors for the baseline, multi-decadal, decadal, and annual versions.

Post-processing baseline product with any of the three density sets improves the baseline product. For SWE values up 130
mm, all three post-processed datasets show similar behaviour, and as the Fig. 7 shows the overestimation of SWE values between 0- and 100-mm present in the baseline retrieval has been mitigated with post-processing. Post-processing also improves the underestimation of large values present in the baseline retrieval, though the improvements are smaller than the improvements for small SWE values.

Post-processing with the annual densities produces worse results than the other two post-processed versions when SWE values up to 500 mm are considered. The worse behaviour of the annual densities for larger SWE values can be caused by the annual density dataset having a larger range of densities than the other two density datasets. If SWE estimation has been close to correct, but the density used in the retrieval is far from the estimated density, post-processing causes SWE estimation to change significantly. A wider range of densities causes more significant changes in the post-processing. Annual and multi-decadal

density sets had higher estimates for large densities than decadal densities, which explains the positive mean error for SWE estimates higher than 150 mm.

**Table 2.** Results of validation for different Eurasian datasets for years 2000-2009, left values are for SWE < 500 mm and bold values are for SWE < 150 mm.

|  | Bias [mm] | RMSE [mm] | MAE [mm] | Correlation coefficient |
|---|---|---|---|---|
| GlobSnow v3.0 (Eurasia) | 2.9/**10.0** | 39.5/**29.7** | 27.2/**23.2** | 0.73/**0.74** |
| Multi-decadal | -1.0/**4.6** | 37.7/**28.0** | 23.9/**20.0** | 0.77/**0.77** |
| Decadal | -2.5/**3.2** | 37.4/**27.5** | 23.6/**19.5** | 0.77/**0.77** |
| Annual | -2.2/**3.7** | 38.5/**27.5** | 23.9/**19.6** | 0.77/**0.78** |

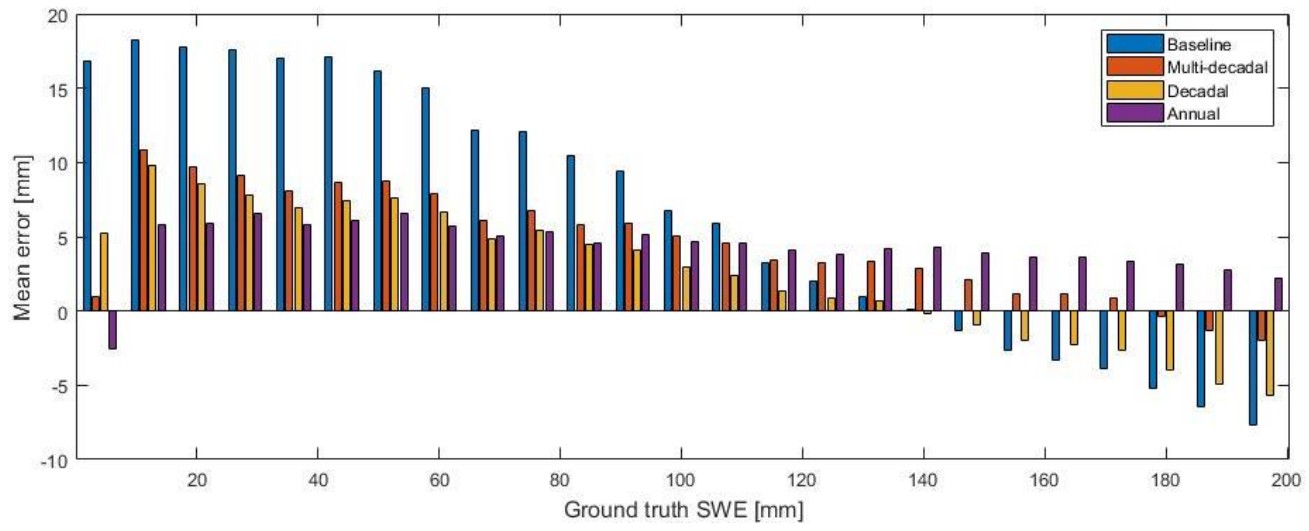


**Figure 7**. The mean error for the baseline, multi-decadal, decadal and annual SWE estimates, 2000-2009 Eurasia.

**3.1 Northern hemisphere, 1979-2018**

Based on the results obtained for the years 2000-2009, the decadal version of snow densities was extended to cover the whole northern hemisphere and the whole period of the baseline retrieval. Four separate sets of density maps were made, each covering one decade starting from the 1980s and ending in the 2010s. The decadal density maps were calculated using static ten-year periods not running ten-year averages because these two methods were found to produce very similar results and producing four set of maps is considerably simpler than producing separate maps for every year using moving ten-year

averages. The density maps were made using the methods explained in chapter 2, with a few exceptions: 1) the spatial interpolation was performed for latitudes from 35° N to 80° N, and for longitudes from -180° W to 180° E to cover the same area as the baseline retrieval and 2) the North American dataset was filtered before adding DOW values. Filtering consisted of two steps. First, locations that were within the same EASE-grid cell were combined and the average value of measurements was calculated for each day. Then mountain mask was applied to remove locations in mountainous areas. After these filtering

steps, the North American implementation set contains 869 location and validation set is made from 201 locations.

These new decadal snow density maps were used to post-process the whole GSv3.0 baseline dataset and the results obtained were compared to validation datasets from Eurasia and North America. Again, SWE values up to 500 mm and 150 mm were validated separately. Validation was performed for the whole winter (September to June) and separately for February, April,

and December. Table 3 summarizes the results of this evaluation. Table 3 shows results for the whole northern hemisphere and separately for Eurasia and North America.




**Table 3.** Results of validation for whole northern hemisphere, Eurasia and North America **for whole winter, February, April, and December** for 1979-2018. Left values are for SWE < 500 mm and bold values are for SWE < 150 mm.

| Area | Period | Product | Bias [mm] | RMSE [mm] | MAE [mm] | Correlation coefficient |
|------|--------|---------|-----------|-----------|----------|-------------------------|
| Northern hemisphere | Winter | GSv3.0 | 1.2/**9.7** | 43.5/**31.6** | 29.3/**24.5** | 0.70/**0.71** |
| | | post-processed | -1.4/**5.4** | 42.5/**30.8** | 27.0/**22.0** | 0.73/**0.73** |
| | December | GSv3.0 | 14.6/**16.1** | 29.5/**25.8** | 21.8/**20.8** | 0.68/**0.75** |
| | | Post-processed | 0.1/**1.7** | 24.3/**18.5** | 14.7/**13.4** | 0.69/**0.75** |
| | February | GSv3.0 | 11.1/**16.8** | 36.9/**30.6** | 26.6/**24.3** | 0.75/**0.77** |
| | | Post-processed | 5.2/**11.2** | 36.4/**28.6** | 24.6/**21.4** | 0.74/**0.76** |
| | April | GSv3.0 | -32.7/**-17.4** | 63.9/**40.8** | 43.8/**31.3** | 0.68/**0.61** |
| | | Post-processed | -20.5/**-9.5** | 61.1/**42.4** | 41.8/**32.3** | 0.68/**0.61** |
| North America | Winter | GSv3.0 | -19.7/**3.3** | 70.9/**42.7** | 48.2/**33.1** | 0.50/**0.48** |
| | | post-processed | -16.0/**4.8** | 69.9/**45.6** | 47.4/**34.1** | 0.53/**0.48** |
| | December | GSv3.0 | 14.8/**18.3** | 41.2/**36.6** | 30.2/**27.6** | 0.51/**0.48** |
| | | Post-processed | 6.0/**10.3** | 38.4/**31.0** | 26.2/**23.1** | 0.48/**0.51** |
| | February | GSv3.0 | -0.7/**14.7** | 51.4/**37.3** | 36.7/**29.7** | 0.67/**0.63** |
| | | Post-processed | -2.4/**12.5** | 52.9/**39.0** | 37.2/**29.7** | 0.65/**0.60** |
| | April | GSv3.0 | -75.2/**-36.9** | 110.8/**60.7** | 83.0/**48.9** | 0.40/**0.32** |
| | | Post-processed | -60.2/**-25.2** | 105.0/**62.3** | 77.8/**49.7** | 0.40/**0.32** |
| Eurasia | Winter | GSv3.0 | 2.5/**10.0** | 41.2/**30.8** | 28.2/**24.0** | 0.73/**0.72** |
| | | post-processed | -0.5/**5.5** | 40.2/**29.8** | 25.8/**21.4** | 0.75/**0.75** |
| | December | GSv3.0 | 11.0/**12.4** | 29.5/**26.1** | 22.0/**21.1** | 0.67/**0.72** |
| | | Post-processed | 0.0/**1.5** | 23.9/**18.1** | 14.5/**13.1** | 0.70/**0.76** |
| | February | GSv3.0 | 10.5/**15.5** | 36.5/**30.9** | 26.5/**24.5** | 0.75/**0.76** |
| | | Post-processed | 5.6/**11.1** | 35.5/**28.1** | 24.0/**21.1** | 0.74/**0.76** |
| | April | GSv3.0 | -30.2/**-16.8** | 59.6/**39.6** | 42.3/**30.4** | 0.72/**0.64** |
| | | Post-processed | -18.3/**-8.8** | 57.6/**41.4** | 39.8/**31.5** | 0.71/**0.63** |

The RMSE was 43.5 mm and 42.5 mm for the baseline retrieval and retrieval post-processed with decadal densities, respectively. Figure 8 shows scatter plots for the whole northern hemisphere for the baseline (a) and the post-processed datasets (b) and as expected, post-processing reduces overestimation of SWE values between 10 and 100 mm. Figure 8 also shows density scatter plots for the baseline and the post-processed datasets for Eurasia ((c), (d)) and North America ((e), (f)) datasets. The post-processed Eurasian dataset shows similar improvements as the overall northern hemisphere dataset, overestimation

of SWE values between 10 and 100 mm has been mitigated. For Eurasia, post-processing improves estimations every month when SWE values up to 500 mm are considered and the biggest improvement in RSME happens in December (over 5 mm). When SWE values only up to 150 mm are considered, we see similar big improvement in December, but in April baseline retrieval produces better results than the post-processed dataset.

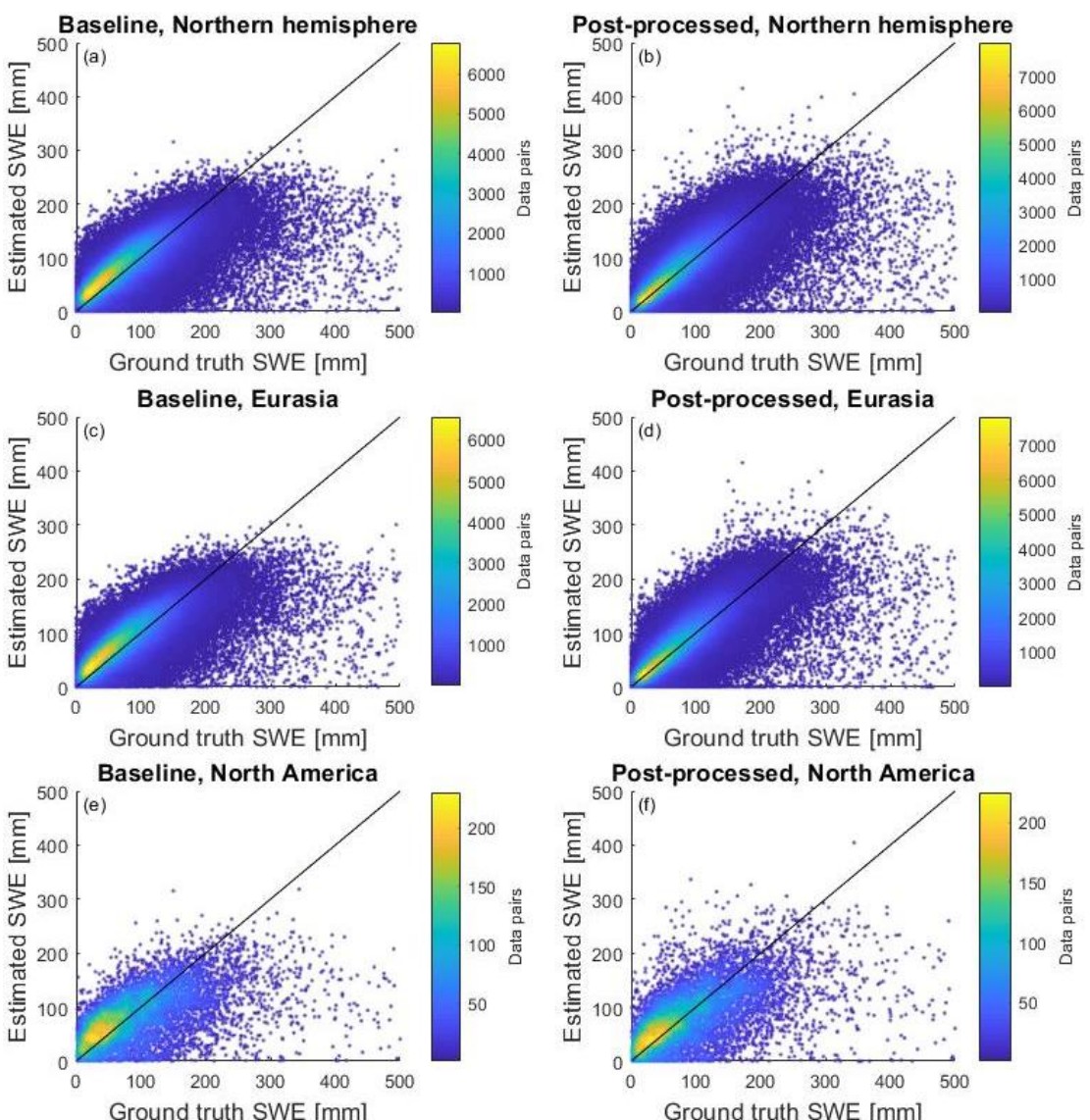


**Figure 8.** Density scatterplots of GSv3.0 baseline and post-processed retrieval accuracy for 1979-2018 for whole northern hemisphere, Eurasia, and North America.

North American datasets also shows some improvements but not as clearly as the Eurasian dataset. For North America SWE
values are improved for December and April when SWE values up to 500 mm are considered. When SWE values up to 150 mm are considered, improvements are seen only in December. Figure 9, which shows the mean retrieval error and standard deviation of the error for baseline and post-processed retrievals for the northern hemisphere, Eurasia, and North America, agrees with these findings. Mean errors for North America are similar for SWE values below 100 mm for the baseline and post-processed retrievals butfor larger values, the mean errors of the post-processed dataset are smaller than the errors of the

baseline set. For Eurasia and northern hemisphere, the mean error is systematically smaller for the post-processed dataset than for the baseline dataset. Figure 10 shows SWE maps for February 6, 2011, for the baseline and post-processed retrievals. Figure 10 shows also difference between these two SWE maps (baseline minus post-processed). Maps show how SWE values are lower in the post-processed maps and the map of differences reveals that post-processing causes bigger changes in Eurasia than in North America.

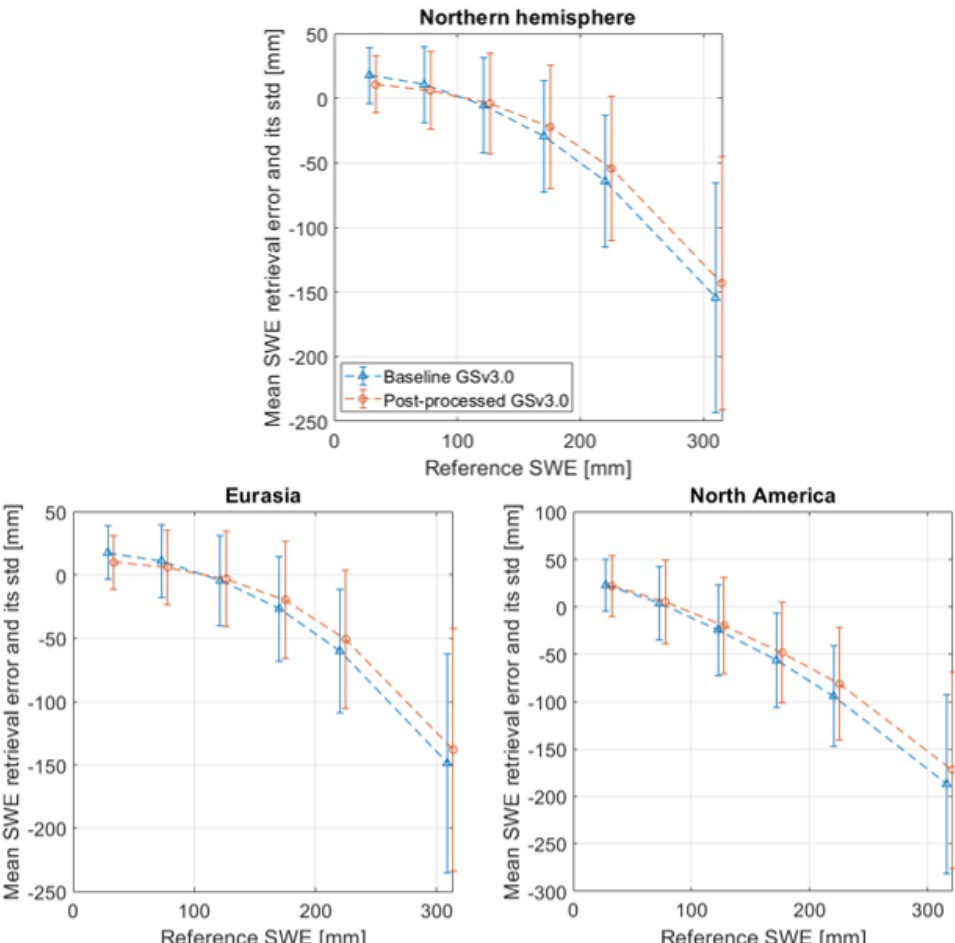


**Figure 9.** Comparisons of the mean error and standard deviations between baseline and post-processed SWE datasets.

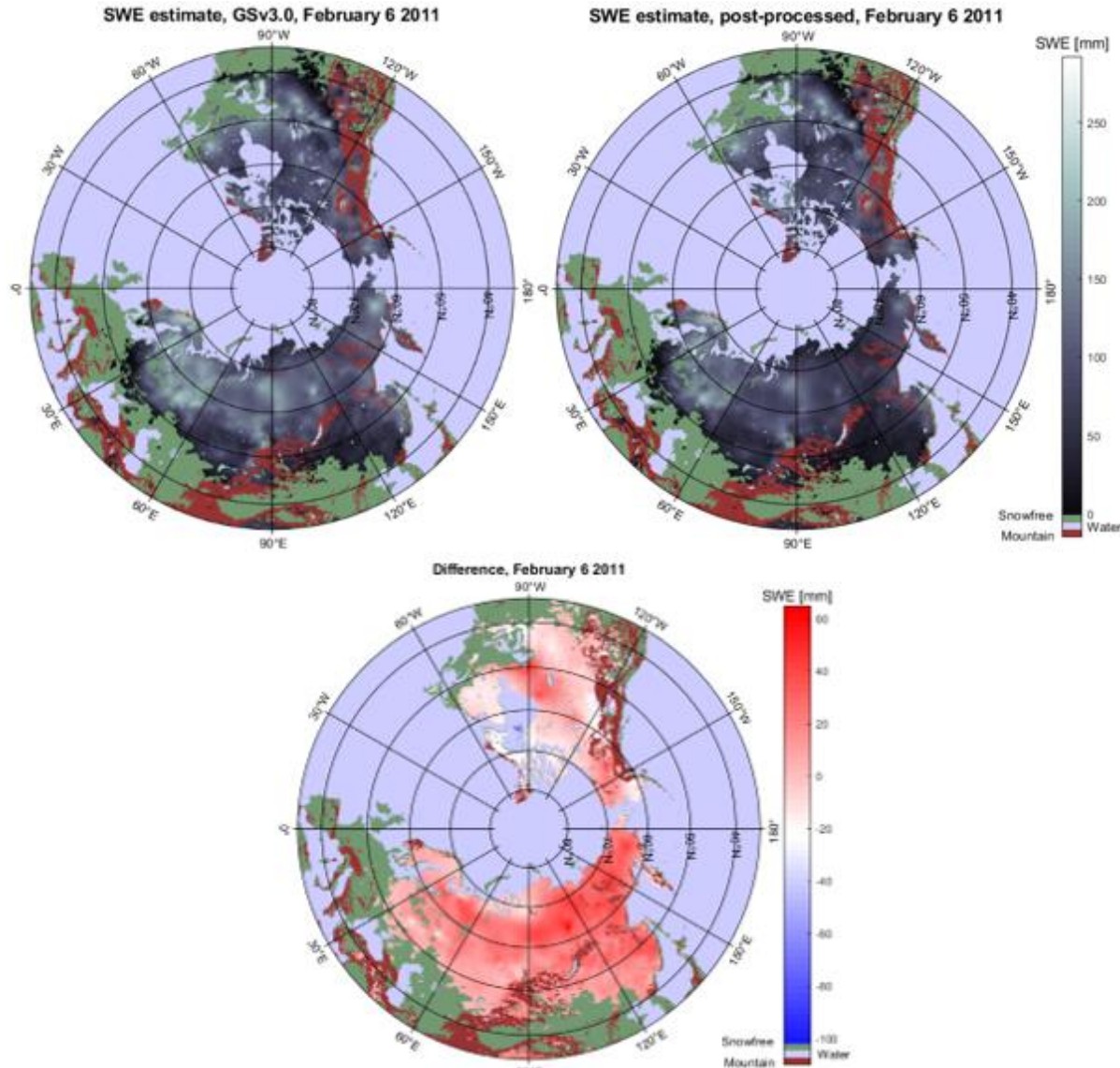

**Figure 10.** SWE maps for baseline GSv3.0 retrieval (top left), post-processed retrieval (top right) and difference between GSv3.0 and post-340 processed retrieval (post-processed subtracted from the baseline) for February 6, 2011. The post-processed SWE values are lower as overestimation of small SWE values is reduced.

## 4 Discussion

GlobSnow 3.0 SWE retrieval accuracy is affected by the overestimation of small SWE values and underestimation of large SWE values. Passive microwave SWE retrievals tend to systematically underestimate SWE under deep snow conditions as 345 the snowpack changes from scattering medium to a source of emission, which becomes significant for SWE retrievals over

about 150 mm. While the GlobSnow retrieval estimates large SWE values better than stand-alone passive microwave SWE retrieval, errors are still evident for deep snow. The underestimation of SWE in GlobSnow retrieval under deep snow conditions is additionally driven by the constant density that relates snow depth and SWE. The constant density tends to decrease SWE estimates for late winter when the snowpacks are usually denser than the constant density applied. Post-processing with dynamic densities improves the overall deep snow retrieval performance as the SWE values are scaled up with larger dynamic snow density. Similarly, the post-processing helps to improve the overestimation of small SWE values. The constant snow density used in the retrieval tends to be too large in the early winter when the snow is fresh, and the density is at its lowest.

Even though the improvements obtained by post-processing for the whole winter are not large (RMSE reduced by about 1 mm), significant improvements are still gained. The small changes in whole dataset are expected, as most of the validation data are from areas where there are no large mistakes in the baseline product and snow density is close to the constant snow density of 240 km m-3 in mid-winter. However, the monthly analyses (table 3) shows that the improvements in overestimation of SWE values in early winter (December) are significant, the RMSE is reduced by about 5 (7) mm for SWE values up to 500 (150) mm and bias is reduced by 15 mm for both versions of validation. Figure 7 also shows that significant improvements (5 -10 mm smaller mean error) are obtained with post-processing for SWE < 100 mm and SWE > 170 mm.

Improvements produced by post-processing are more significant for the Eurasia than for the North America. The North American reference measurements consist of both snow transect and point measurements which may affect the results. The measurement frequency of the daily SNOTEL data differ also from the Eurasia and Canadian data which have measurements from every 10 to 15 days. However, when SNOTEL data was resampled to 10-day intervals and validation parameters were recalculated, no significant differences were detected. Other possible source of error in the SNOTEL data is the location of snow depth sensor. In some locations, snow depth is not measured on top of the snow pillow but next to the pillow and this may affect the accuracy of the snow density values. However, it should be noted that the GlobSnow retrieval is known to have worse performance in Canada and this is partly due to higher average SWE compared to Eurasia (Mortimer et al., 2020).

As shown by the results, post-processing with dynamic snow densities can be used to improve existing datasets and the post-processing procedure is straightforward to perform. However, implementing dynamic snow densities directly into the retrieval could possibly produce even more notable improvements and is a key research topic to be studied in the future. Snow density is one of the input parameters of the HUT snow emission model, determining the absorption coefficient in snow, refraction, and transmissivity at the air-snow interface and transmissivity at the snow-ground interface through modelled permittivity of the snow layer (Pulliainen et al., 1999). The HUT model is used for determining effective snow grain size over weather station locations as well as for obtaining the final SWE estimates using numeric model inversion (Takala et al. 2011). The final snow grain size (and its variance) at each location is the average grain size of six nearest stations. If the true snow density between stations changes significantly the variance of the estimated snow grain sizes increases. This in turn potentially reduces the

weight of radiometer measurements on the final SWE estimation, as well as the accuracy of the individual grain size estimates. Similarly, applying a wrong density value in the final retrieval step potentially deteriorates the accuracy of the HUT model and thus retrieval skill.

Although better results might be achieved with implementing dynamic densities into the retrieval, post-processing is justified
as it can be used to study different implementations of the snow densities with relative ease and the results obtained with post-processing are similar to results obtained with implementing dynamic densities in retrieval. Running the full retrieval algorithm is very time consuming, and as such not well suited for testing small changes in methodology. Post-processing can be used to study which densities and methodologies produce the best results and these densities can then be implemented into a final retrieval product. Also, as some areas have more snow density information available than others, the introduced post-
processing methods provide feasible tools to improve the accuracy of the global GlobSnow data based SWE estimates for regional hydrological applications in such areas where snow density information is available.

Different approaches for varying snow density for satellite based SWE retrievals have been used. The AMSR-E v1.0 product (Kelly, 2009) uses spatially varying but temporally static snow density maps based on snow classes suggested in Liston et al.
(1998). However, evaluation of this product by Tedesco and Narvekar (2010) pointed out the need to have also temporal variability in the snow density. The AMSR-E SWE v2.0 product uses snow spatially and temporally varying density maps based on Strum et al. (2010) for converting snow depth to SWE (Tedesco et al., 2016). However, the densities based Strum et al. (2010) cause large overestimation of small SWE value when used for post-processing the GlobSnow product as seen in Fig. 11, which shows the mean retrieval error for GSv3.0 SWE values post-processed with densities based on Strum et al. (2010)
method for Eurasia for 2000-2009.

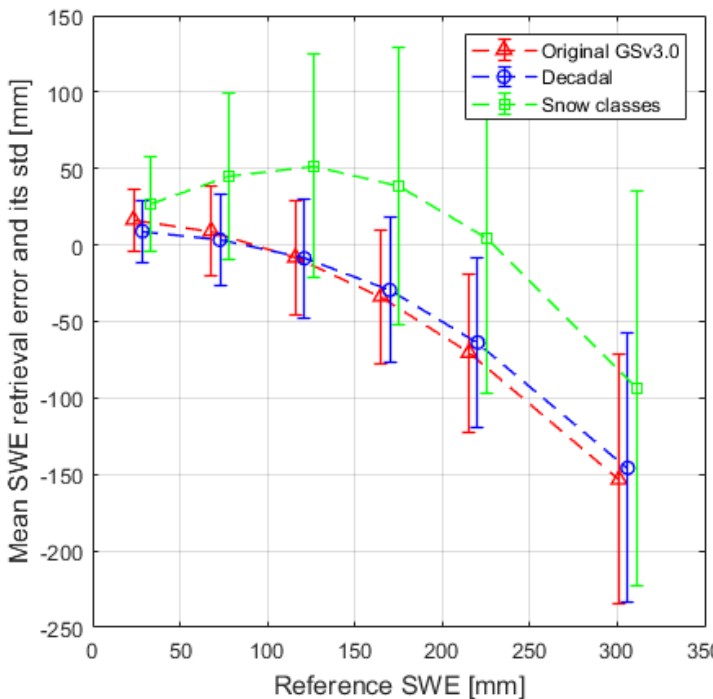

**Figure 11.** Comparisons of the mean error and standard deviations between baseline and datasets post-processed with decadal snow densities and densities based on fixed snow classes according to Sturm et al. (2010).


In this research, linear interpolation was used for temporal interpolation to obtain density measurements for days without any measurements. However, snow density measurements may contain some errors and these errors can influence the results of the linear interpolation. Thus, alternative interpolation methods for determining behaviour of snow density throughout the snow season, such as higher degree polynomial interpolation of averaged values or yearly values, could be evaluated in future
investigations.

## 5 Conclusions

In this study spatially and temporally changing snow density fields were implemented using snow density measurements from Eurasia and North America. The dynamic snow density fields were used to post-process GlobSnow version 3.0 SWE retrievals. Post-processing was found to improve overestimation of small SWE values between 10-100 mm and underestimation of large
SWE values. This indicates that the constant density used in the baseline retrieval is too large for early winter and too small for late winter. The overall results indicate a clear path forward to improve the overall GlobSnow SWE retrieval methodology by application of dynamic snow density in the post-processing scheme.


**Code and data availability.** The GlobSnow code is available at: http://www.globsnow.info/swe/archive_v3.0/source_codes/ the GlobSnow v3.0 data is available at: https://www.globsnow.info/swe/archive_v3.0/L3A_daily_SWE/. The snow density processing code is available upon request from corresponding author.

**Author contribution**. P.V., K.L., J.L and J.P. conceived the concept of the study; P.V. performed the analyses, data processing, computing and produced the first draft of the manuscript, which was subsequently edited by K.L, J.L, and J.P.; M.T. and M.M. contributed to the analytical tools and methods.

**Competing interests**. The authors declare that they have no conflict of interest.

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
