# Peer review of "Implementation of dynamic snow density within GlobSnow snow water equivalent retrieval methodology"

_The Cryosphere, 2021_

## Author Response (AR1)

**Response to referee 1**

We thank the referee for his/her valuable comments on the manuscript. Author response to the comments and the changes we plan to make in the revised manuscript are detailed below. Author remarks are written in red, additions to manuscript are written in green. Reviewer remarks are written in black.

**General Comments:**

This manuscript is fairly straightforward and logical. It is mostly well written, but some sections could use clearer language or additional explanation. The study seems worthwhile, because there is a large amount of uncertainty in global snow estimates and assuming a constant snow density over time and space (as in GlobSnow) is not realistic, so any method that could introduce empirical density estimates would be beneficial. However, the improvements in SWE estimation did not seem all that significant/impressive, so I wonder if post-processing is not the best approach. I have some additional specific comments below about the methodology, interpretations, and presentation.

**Specific Comments:**

The title is a bit misleading, because it makes it sound like the dynamic snow density is incorporated directly into the retrieval, when in fact it is applied in post-processing.

We appreciate the reviewer's comment on this. However, we feel that changing the tile of the article is not necessary. For the end-user, there is no difference where the new dynamic density information is injected in the overall processing chain. The key point is that the dynamic snow density has been applied for the GlobSnow product in the retrieval framework and this provides a significantly improved product for the end users. Approach to include it during the assimilation phase is being studied but for the time being the presented approach has produced the best results and will be the baseline for the next GlobSnow product version.

Line 64: Do the authors have a citation that they can provide to show that the Sturm et al. approach did not improve retrieval skill?

Citation to relevant report will be added (line 64):

Luojus, K., Pulliainen, J., Takala, M., Lemmetuinen, J., Kangwa, M., Smolander, T., Cohen, J., Derksen, C.: Preliminary SWE validation report, European Space Agency Study Contract report, 2013.

Lines 141-142: Why was there so much more validation data than implementation data in the Eurasian portion of the analysis, but the opposite in North America?

Large amount of validation data was wanted for Eurasia because different versions of the snow density fields were tested only in Eurasia. We also wanted to use independent SYKE dataset from Finland for validation to see if there were clear biases in the RIHMI-WDC data, which was used both for validation and implementation.

The North American implementation dataset contained 869 sites after all the filtering steps and not all these sites were producing data already in 1979. Thus, the North American implementation set contains only 681 sites for 1979-1989 clustered mostly in few parts of the continent and if there were much less sites available implementing dynamic densities for the whole continent would become challenging.

Line 163: The method does not produce SWE estimates for mountainous areas, but most of the SNOTEL data come from mountainous regions. How do the authors justify using mountain snow to determine the density for snow that is not in the mountains?

SNOTEL sites located in areas for which the SWE is not estimated were removed from the North American implementation dataset. This filtering step was not mentioned in the article by mistake but will be added to revosed version (lines 302-306):

Filtering consisted of two steps. First, locations that were within the same EASE-grid cell were combined and the average value of measurements was calculated for each day. Then mountain mask was applied to remove locations in mountainous areas. After these filtering steps, the North American implementation set contains 869 location and validation set is made from 201 locations.

Figure 5: When the authors compare the in-situ densities used to reprocess GlobSnow to their nearest neighbors (used for ground truth), is a similar pattern seen, or does it better fit the 1:1 line? Is the interpolation smearing out the range/variability of the density values? If so, the authors might search for a different interpolation method. Additionally, by interpolating across large distances (which may have different land cover, elevation, etc.), the authors are effectively taking local density estimates assuming that they are representative of large areas. Do the authors think this is a fair thing to do? Might it be better to use an interpolation method akin to PRISM (Daly et al.) to take land surface properties into account?

We are using kriging interpolation to form a continuous map of the snow densities. Utilization of a more complex interpolation scheme (than the kriging interpolation) could be a potential avenue for future investigations, but the current agreement between the interpolated snow density maps and independent densities (the densities from the validation dataset) is a rather good. Using nearest neighbor interpolation would probably be quite problematic, and therefore we have not applied that. Further, we know that the variability of densities may be quite a bit higher in mountainous regions, so those are masked out, i.e. pixels where standard deviation of elevation exceeds a 200 m threshold are masked out from the consideration.

When comparing in-situ implementation snow densies and their nearest validation densies similar patter to the pattern of the figure 5 (comparison of interpolated densies to validation in-situ data) can be seen (see Fig. 1 below). This indicates that kriging interpolation is not smearing out the variability of the density values significantly. Most snow density values range between 150 kg m$^{-3}$ and 350 kg m$^{-3}$, which can explain the departure from 1:1 fit for small and large density values.

[Figure]

**Figure 1**. Left: Comparison of in-situ implementation densities and their nearest neighbors. Right: Comparison of interpolated densities and ground truth densities at validation sites (figure 5 from article).

Line 244: "…SWE values up to 500 mm and SWE values up to 150 mm…". This is a confusing sentence, and I was only able to understand what the authors meant by looking at Table 2.

This will be fixed with new sentences (lines 253-254):

Two validation were performed, the first validation took into account SWE values up to 500 mm and the second validation considered SWE values only up to 150 mm, as the bulk of the observations are below this value

Line 269: Is GlobSnow calibrated to SWE in any way? If so, then post-processing might not be a good idea? The authors note in lines 338-349 that implementing densities directly into the retrieval may be more beneficial, as a wrong density may lead to reduced retrieval skill. So, then why bother with post-processing?

The GlobSnow 3.0 product used here is not calibrated to SWE (e.g. SWE measured by the snow courses) in any way. The retrieval process does, however, assimilate in-situ measured snow depths from weather stations in the retrieval. In the original scheme these snow depths are converted to SWE using a constant value for density (240 kg/m3), and replacing this value in post processing is actually a fairly straightforward step, and a necessary step, as shown by the results in this manuscript. We thus feel that the improvements obtained with post-processing are significant, in particular for low values of SWE.

A small error is introduced in using only post-processing since simulation of the snow absorption coefficient in inversion of the HUT forward model is still done with the original constant value for density (see e.g. Pulliainen et al., 1999, Pulliainen 2006). However, post-processing is justified as it can be used to study different implementations of the snow densities with relative ease and the results obtained with post-processing are similar to results obtained with implementing dynamic densities in retrieval. Running the full retrieval algorithm is very time consuming, and as such not well suited for testing small changes in methodology. Post-processing can be used to study which densities and methodologies produce the best results and these densities can then be implemented into a final retrieval product.

Also, as some areas have more snow density information available than others, the introduced post-processing methods provide feasible tools to improve the accuracy of the global GlobSnow data based SWE estimates for regional hydrological applications in such areas where snow density information is available.

Discussion about using post-processing will be added to discussion chapter (lines 416-424):

Although better results might be achieved with implementing dynamic densities into the retrieval, post-processing is justified as it can be used to study different implementations of the snow densities with relative ease and the results obtained with post-processing are similar to results obtained with implementing dynamic densities in retrieval. Running the full retrieval algorithm is time consuming, and as such not well suited for testing small changes in methodology. Post-processing can be used to study which densities and methodologies produce the best results and these densities can then be implemented into a final retrieval product. Also, as some areas have more snow density information available than others, the introduced post-processing methods provide feasible tools to improve the accuracy of the global GlobSnow data based SWE estimates for regional hydrological applications in such areas where snow density information is available.

Line 272: Calculating decadal density maps and applying them to the entire decade seems a bit arbitrary. For instance, the density applied to data from 1981 would be from (approximately) the next 10 years into the future, while for 1989 the density would be from (approximately) the past 10 years. Why not calculate a running decadal average, centered on the year of interest?

A centred decadal average was used because it produces very similar results to a running decadal average but is considerably simpler to produce. The validation parameters for the running decadal average (centred decadal average) snow densities are RMSE = 48.9 (48.9), bias = 0.49 (0.88) and correlation coefficient= 0.71 (0.71) for Eurasia for years 2000-2009. Discussion about using centred decadal averages will be added to the article (lines 297-300).

The decadal density maps were calculated using static ten-year periods not running ten-year averages because these two methods were found to produce very similar results and producing four set of maps is considerably simpler than producing separate maps for every year using moving ten-year averages.

Line 277-278: I don't know what the authors mean in this sentence.

This sentence was aiming to explain that the calculated validation parameters for the decadal densities over the longer period of 1979-2018 are similar to the validation parameters over the shorter period of 2000-2009. However, the sentence is probably unnecessary and will be removed as the validation parameter are listed in the next sentence.

Table 3: The improvements in post-processed SWE seem marginal, so I wonder if post-processing is worth doing at all. Do the authors have any way to evaluate the statistical significance of the improvements in SWE?

Large improvements in the whole dataset are not expected as most of the validation data are from areas where there are no large mistakes in the baseline product and snow density is close to the constant snow density of 240 km m$^{-3}$ in mid-winter. To highlight the improvements obtained, we have added analyses of specific months to table 3 (shown below). Monthly analyses show, for example, that improvements of 5 mm for SWE values up to 500 mm and 7 mm for values up to 150 mm were obtained for December. Significant improvements are also observed in bias. Figure 2 below (fixed from figure 6 present in the article) also shows that significant improvements (5 -10 mm smaller mean error) are obtained with post-processing for SWE < 100 mm and SWE > 170 mm.

We can also study the statistical differences between the baseline and the post-processed datasets with T-test. The p-value is 3.48E-43 when the T-test is performed for the post-processed and the baseline differences between measured and estimated SWE values. The small p-value indicates that the post-processed and baseline sets differ significantly from each other.

We will add discussion about this to article (lines 387-394):

Even though the improvements obtained by post-processing for the whole winter are not large (RMSE reduced by about 1 mm), significant improvements are still gained. The small changes in whole dataset are expected, as most of the validation data are from areas where there are no large mistakes in the baseline product and snow density is close to the constant snow density of 240 km m$^{-3}$ in mid-winter. However, the monthly analyses (table 3) shows that the improvements in overestimation of SWE values in early winter (December) are significant, the RMSE is reduced by about 5 (7) mm for SWE values up to 500 (150) mm and bias is reduced by 15 mm for both versions of validation. Figure 7 also shows that significant improvements (5 -10 mm smaller mean error) are obtained with post-processing for SWE < 100 mm and SWE > 170 mm.

[Figure]

**Figure 2**. The mean error for the baseline, multi-decadal, decadal, and annual SWE estimates, 2000-2009 Eurasia.

**Table 3.** Results of validation for whole northern hemisphere, North America, and Eurasia for whole winter, February, April, and December for 1979-2018. Left values are for SWE < 500 mm and bold values are for SWE < 150 mm.

| Area | Period | Product | Bias [mm] | RMSE [mm] | MAE [mm] | Correlation coefficient |
|---|---|---|---|---|---|---|
| Northern hemisphere | Winter | GSv3.0 | 1.5/**9.8** | 43.0/**31.3** | 29.0/**24.3** | 0.71/**0.71** |
| | | post-processed | -1.2/**5.4** | 42.0/**30.5** | 26.7/**21.9** | 0.74/**0.74** |
| | December | GSv3.0 | 14.6/**16.1** | 29.5/**25.8** | 21.8/**20.8** | 0.68/**0.75** |
| | | Post-processed | 0.1/**1.7** | 24.3/**18.5** | 14.7/**13.4** | 0.69/**0.75** |
| | February | GSv3.0 | 11.1/**16.8** | 36.9/**30.6** | 26.6/**24.3** | 0.75/**0.77** |
| | | Post-processed | 5.2/**11.2** | 36.4/**28.6** | 24.6/**21.4** | 0.74/**0.76** |
| | April | GSv3.0 | -32.7/**-17.4** | 63.9/**40.8** | 43.8/**31.3** | 0.68/**0.61** |
| | | Post-processed | -20.5/**-9.5** | 61.1/**42.4** | 41.8/**32.3** | 0.68/**0.61** |
| North America | Winter | GSv3.0 | -21.1/**3.1** | 72.4/**42.3** | 48.6/**32.7** | 0.50/**0.49** |
| | | post-processed | -17.3/**4.7** | 71.3/**45.3** | 47.8/**33.7** | 0.53/**0.49** |
| | December | GSv3.0 | 14.8/**18.3** | 41.2/**36.6** | 30.2/**27.6** | 0.51/**0.48** |
| | | Post-processed | 6.0/**10.3** | 38.4/**31.0** | 26.2/**23.1** | 0.48/**0.51** |
| | February | GSv3.0 | -0.7/**14.7** | 51.4/**37.3** | 36.7/**29.7** | 0.67/**0.63** |
| | | Post-processed | -2.4/**12.5** | 52.9/**39.0** | 37.2/**29.7** | 0.65/**0.60** |
| | April | GSv3.0 | -75.2/**-36.9** | 110.8/**60.7** | 83.0/**48.9** | 0.40/**0.32** |
| | | Post-processed | -60.2/**-25.2** | 105.0/**62.3** | 77.8/**49.7** | 0.40/**0.32** |
| Eurasia | Winter | GSv3.0 | 2.5/**10.0** | 41.2/**30.8** | 28.2/**24.0** | 0.73/**0.72** |
| | | post-processed | -0.5/**5.5** | 40.2/**29.8** | 25.8/**21.4** | 0.75/**0.75** |
| | December | GSv3.0 | 11.0/**12.4** | 29.5/**26.1** | 22.0/**21.1** | 0.67/**0.72** |
| | | Post-processed | 0.0/**1.5** | 23.9/**18.1** | 14.5/**13.1** | 0.70/**0.76** |
| | February | GSv3.0 | 10.5/**15.5** | 36.5/**30.9** | 26.5/**24.5** | 0.75/**0.76** |
| | | Post-processed | 5.6/**11.1** | 35.5/**28.1** | 24.0/**21.1** | 0.74/**0.76** |
| | April | GSv3.0 | -30.2/**-16.8** | 59.6/**39.6** | 42.3/**30.4** | 0.72/**0.64** |
| | | Post-processed | -18.3/**-8.8** | 57.6/**41.4** | 39.8/**31.5** | 0.71/**0.63** |

Figure 8: What does the colorbar in this figure show? It is not labelled.

Colorbar shows number of points within a certain area. Label will be added to the figure (see figure 8 below).

Figures 8-10: Why are these 3 separate figures? Why not one figure with 6 different panels?

These 3 figures will be combined into one figure with 6 panels, see below.

[Figure]

**Figure 8.** Density scatterplots of GSv3.0 baseline and post-processed retrieval accuracy for 1979-2018 for whole northern hemisphere, Eurasia, and North America.

Lines 339-347: Do the authors have a citation(s) for this?

Relevant citations will be added (Pulliainen et al., 1999 and Takala et al. 2011).

Lines 350-351: To what methods are the authors referring? This paragraph needs to be further explained.

We are referring to alternative interpolation methods. Additional information will be added to the chapter (lines 439-443):

In this research, linear interpolation was used for temporal interpolation to obtain snow density values for days without any measurements. However, snow density measurements may contain errors, and these errors can influence the results of the linear interpolation. Thus, alternative interpolation methods for determining behaviour of the snow density throughout the snow season, such as higher degree polynomial interpolation of averaged values or yearly values, could be evaluated in future investigations.

**Technical Corrections**

Line 17: "was found to produce the best results"?

Will be corrected.

Lines 58-59: Snow also undergoes metamorphism that can cause density to increase.

Mention of metamorphism will be included to the section (line 59).

Line 130: "The North American dataset"?

Line 133: "The SNOTEL dataset"?

Caption of Figure 4: "for snow transects"?

Line 231: "differs from the other two versions more" – could improve grammar

Figure 8, y-axis label: "Estimated SWE"?

Edits mentioned above will be included, thanks.

Figures 8 & 9: The font sizes of the titles of these two figures are different.

These figures will be combined, and font sizes will be fixed.

**Response to referee 2**

We thank the referee for his/her valuable comments on the manuscript. Author response to the comments and the changes we plan to make in the revised manuscript are detailed below. Author remarks are written in red, additions to manuscript are written in green. Reviewer remarks are written with normal font.

**SUMMARY OF THE PAPER**

This study concerns the treatment of snow density in the GlobSnow SWE product (post-processing only). It develops and tests new dynamic snow density estimates based on data collected at snow courses and telemetered snow sites in the Northern Hemisphere. These estimates are compared to the existing approach, which assumes constant snow density in space and time. The new dynamic approach is first developed at a subset of measurement location. Snow density is averaged each day of winter over different periods (multi-decadal, decadal, and annual), then interpolated in time and spatially mapped using ordinary kriging interpolation. All three approaches show improved mapping of snow density against observations relative to the constant density assumption, however lower density values (300 kg/m3) are underestimated. The snow density estimates based on annual values are slightly improved over the multi-decadal and decadal versions. Post-processing of the GlobSnow SWE data is then conducted for each of the three dynamic density datasets, with SWE errors lowest for the multi-decadal and decadal values of snow density, and improved over the case of constant density. The decadal version was then applied to the northern Hemisphere GlobSnow and SWE was post-processed. This yielded slightly better agreement in SWE compared to ground truth data. SWE errors were consistently reduced at all reference SWE values in Eurasia but only at SWE greater than 100 mm in North America.

**COMMENTS**

1. The Sturm density method was briefly mentioned as a candidate approach, but found to not improve retrieval skill notably (Lines 63-64). Was that analysis published? If not, it could be worth including in the current study, as it would be an interesting point of comparison that would be of interest to the community.

Citation to relevant report will be added (line 64). Also, sort comparison of post-processing with the Strum density method will be added to article (see answer to comment #4).

Luojus, K., Pulliainen, J., Takala, M., Lemmetyinen, J., Kangwa, M., Smolander, T., Cohen, J., Derksen, C.: Preliminary SWE validation report, European Space Agency Study Contract report, 2013.

2. For the SNOTEL data, did the authors check to ensure that the snow depth sensor measures snow depth on top of the snow pillow? This is not always the case at SNOTEL stations, and when snow depth is recorded at a location next to the snow pillow, an estimate of density from the SWE and depth data may be biased. Unfortunately, documentation on the layout of sites is inconsistent. Careful inspection of site photos can reveal which SNOTEL stations may have collocated SWE and depth measurements.

We did not check that snow depth is measured on top of the snow pillow and this could potentially affect the results. However, the uncertainties of the SNOTEL data are, in any case, much larger compared to the Eurasian data and this is mentioned in the article. However, in order to come up with a dynamic snow

density estimates for the North American sector, this SNOTEL dataset is the best one to our knowledge (in regard to temporal coverage, spatial coverage and its applicability for the task) and it is therefore applied here. Additionally, in cases where snow depth is not measured in top of the snow pillow, it is measured close by the pillow and the bias this causes should not be very large. We will add discussion about SNOTEL data to the article (lines 395-402), see answer to comment #5.

3. The results in Figures 5 and 6 appear to be contradictory. Figure 5 suggests that that snow density estimation is improved (but only slightly) relative to decadal and multi-decadal across a range of ground truth density values. In contrast, Figure 6 suggests much larger SWE errors associated with the annually-derived snow density when compared to multi-decadal and decadal versions. Given the post-processing (equation 5), the differences in SWE must be due solely to differences in snow density errors and not errors in baseline SWE. However, Figure 5 suggests the snow density errors are lower for the annual estimate. How do you rectify these conflicting results? Are you sure these have been analyzed correctly?

There was a mistake in figure 6, which explains the conflicting results. Fixed figure below. Most of the 3.1.2 has been rewritten according to corrected results (lines 250-271).

Post-processing baseline product with any of the three density sets improves the baseline product. For SWE values up 130 mm, all three post-processed datasets show similar behaviour, and as the Fig. 7 shows the overestimation of SWE values between 0- and 100-mm present in the baseline retrieval has been mitigated with post-processing. Post-processing also improves the underestimation of large values present in the baseline retrieval, though the improvements are smaller than the improvements for small SWE values.

Post-processing with the annual densities produces worse results than the other two post-processed versions when SWE values up to 500 mm are considered. The worse behaviour of the annual densities for larger SWE values can be caused by the annual density dataset having a larger range of densities than the other two density datasets. If SWE estimation has been close to correct, but the density used in the retrieval is far from the estimated density, post-processing causes SWE estimation to change significantly. A wider range of densities causes more significant changes in the post-processing. Annual and multi-decadal density sets had higher estimates for large densities than decadal densities, which explains the positive mean error for SWE estimates higher than 150 mm.

[Figure]

**Figure 6**. The mean error for the baseline, multi-decadal, decadal, and annual SWE estimates, 2000-2009 Eurasia.

4. Overall, the discussion section is brief and does little to connect the current study to previous work (no other studies are actually cited in the discussion). For example, there have been other studies that have compared the constant density assumption versus dynamic snow density for passive microwave SWE retrievals and found improvements with dynamic snow density (e.g., Tedesco and Narvekar, 2010), and other studies have called for improved density representation for passive microwave (Kelly et al., 2003). How do the results of the current study build on established knowledge?

Discussion section will be expanded (see also answers to comments #5 and #6) and following section about different dynamic snow densities will be added (lines 426-434):

Different approaches for varying snow density for satellite based SWE retrievals have been used. The AMSR-E v1.0 product (Kelly, 2009) uses spatially varying but temporally static snow density maps based on snow classes suggested in Liston et al. (1998). However, evaluation of this product by Tedesco and Narvekar (2010) pointed out the need to have also temporal variability in the snow density. The AMSR-E SWE v2.0 product uses snow spatially and temporally varying density maps based on Strum et al. (2010) for converting snow depth to SWE (Tedesco et al., 2016). However, the densities based Strum et al. (2010) cause large overestimation of small SWE value when used for post-processing the GlobSnow product as seen in Fig. 11, which shows the mean retrieval error for GSv3.0 SWE values post-processed with densities based on Strum et al. (2010) method for Eurasia for 2000-2009.

[Figure]

**Figure 11.** Comparisons of the mean error and standard deviations between baseline and datasets post-processed with decadal snow densities and densities based on fixed snow classes according to Sturm et al. (2010).

5. Ignoring the potential issues with SNOTEL-based values snow density (see comment #2 above), there are issues with the location of SNOTEL stations (mountains where snow is deeper and possibly denser, and GlobSnow does not produce SWE), and differences in temporal sampling. To the latter concern, it would be possible to check whether the temporal sampling of snow density influences the density curve development and validation, compared to the case of less frequent snow course measurements (e.g., Canada and Eurasia). The differences in GlobSnow SWE for constant versus dynamic snow density were smaller in North America (where SNOTEL are located) versus Eurasia (snow courses only), as seen in Figure

7. To what degree do the differences in daily snow density (SNOTEL) versus 10-15 day snow density (elsewhere) influence the result? This could be checked by sampling the SNOTEL density to a similar interval as snow courses and recomputing the statistics.

SNOTEL sites located in areas for which the SWE is not estimated were removed from the North American implementation dataset. This filtering step was not mentioned in the article by mistake but will be added to the new version (lines 302-306).

Filtering consisted of two steps. First, locations that were within the same EASE-grid cell were combined and the average value of measurements was calculated for each day. Then mountain mask was applied to remove locations in mountainous areas. After these filtering steps, the North American implementation set contains 869 location and validation set is made from 201 locations.

Validation statistics were recomputed using resampled SNOTEL data, but no significant changes were detected. However, the worse performance of the GlobSnow SWE retrieval over Canada (and North America) is expected and partly due to larger higher average SWE compared to Eurasia (Mortimer et al. 2020; Luojus et al. 2021). We will add discussion about this to article (lines 395-403):

Improvements produced by post-processing are more significant for the Eurasia than for the North America. The North American reference measurements consist of both snow transect and point measurements which may affect the results. The measurement frequency of the daily SNOTEL data differ also from the Eurasia and Canadian data which have measurements from every 10 to 15 days. However, when SNOTEL data was resampled to 10-day intervals and validation parameters were recalculated, no significant differences were detected. Other possible source of error in the SNOTEL data is the location of snow depth sensor. In some locations, snow depth is not measured on top of the snow pillow but next to the pillow and this may affect the accuracy of the snow density values. However, it should be noted that the GlobSnow retrieval is known to have worse performance in Canada and this is partly due to higher average SWE compared to Eurasia (Mortimer et al., 2020).

6. The results seem rather marginal and potentially overstated. An example is Lines 292-293, which state 'the accuracy is considerably better with RMSE for the baseline being 31.87 mm and for the post-processed dataset, the error is 30.87 mm' I question the significance of a 1 mm reduction in SWE error and whether this is 'considerably better'.

We feel that improvements are significant because the global dataset is large and big changes in the RSME are unlikely. Most validation points are in areas where there are no large mistakes in the baseline product but when looking at figures 8-10 or 11 clear improvements are still visible. Fixed figure 6 also shows that significant improvements (5 -10 mm smaller mean error) are obtained with post-processing for SWE < 100 mm and SWE > 170 mm.

To show some of these improvements more clearly, we have added analyses of specific months to table 3 (line 315). Monthly analyses show, for example, that the improvement in SWE was 5 mm for SWE values up to 500 mm and 7 mm for values up to 150 mm for December. Monthly analysis also reveals significant improvements in bias. The bias is reduced by over 10 mm for Eurasia and about 8 mm for North America for December.

We will add discussion about this to article (lines 387-394):

Even though the improvements obtained by post-processing for the whole winter are not large (RMSE reduced by about 1 mm), significant improvements are still gained. The small changes in whole dataset are expected, as most of the validation data are from areas where there are no large mistakes in the baseline product and snow density is close to the constant snow density of 240 km m$^{-3}$ in mid-winter. However, the monthly analyses (table 3) shows that the improvements in overestimation of SWE values in early winter (December) are significant, the RMSE is reduced by about 5 (7) mm for SWE values up to 500 (150) mm and bias is reduced by 15 mm for both versions of validation. Figure 7 also shows that significant improvements (5 -10 mm smaller mean error) are obtained with post-processing for SWE < 100 mm and SWE > 170 mm.

**Table 3.** Results of validation for whole northern hemisphere, North America, and Eurasia for whole winter, February, April, and December for 1979-2018. Left values are for SWE < 500 mm and bold values are for SWE < 150 mm.

| Area | Period | Product | Bias [mm] | RMSE [mm] | MAE [mm] | Correlation coefficient |
|---|---|---|---|---|---|---|
| | Winter | GSv3.0 | 1.5/**9.8** | 43.0/**31.3** | 29.0/**24.3** | 0.71/**0.71** |
| | | post-processed | -1.2/**5.4** | 42.0/**30.5** | 26.7/**21.9** | 0.74/**0.74** |
| | December | GSv3.0 | 14.6/**16.1** | 29.5/**25.8** | 21.8/**20.8** | 0.68/**0.75** |
| Northern | | Post-processed | 0.1/**1.7** | 24.3/**18.5** | 14.7/**13.4** | 0.69/**0.75** |
| hemisphere | February | GSv3.0 | 11.1/**16.8** | 36.9/**30.6** | 26.6/**24.3** | 0.75/**0.77** |
| | | Post-processed | 5.2/**11.2** | 36.4/**28.6** | 24.6/**21.4** | 0.74/**0.76** |
| | April | GSv3.0 | -32.7/**-17.4** | 63.9/**40.8** | 43.8/**31.3** | 0.68/**0.61** |
| | | Post-processed | -20.5/**-9.5** | 61.1/**42.4** | 41.8/**32.3** | 0.68/**0.61** |
| | | | | | | |
| | Winter | GSv3.0 | -21.1/**3.1** | 72.4/**42.3** | 48.6/**32.7** | 0.50/**0.49** |
| | | post-processed | -17.3/**4.7** | 71.3/**45.3** | 47.8/**33.7** | 0.53/**0.49** |
| | December | GSv3.0 | 14.8/**18.3** | 41.2/**36.6** | 30.2/**27.6** | 0.51/**0.48** |
| North | | Post-processed | 6.0/**10.3** | 38.4/**31.0** | 26.2/**23.1** | 0.48/**0.51** |
| America | February | GSv3.0 | -0.7/**14.7** | 51.4/**37.3** | 36.7/**29.7** | 0.67/**0.63** |
| | | Post-processed | -2.4/**12.5** | 52.9/**39.0** | 37.2/**29.7** | 0.65/**0.60** |
| | April | GSv3.0 | -75.2/**-36.9** | 110.8/**60.7** | 83.0/**48.9** | 0.40/**0.32** |
| | | Post-processed | -60.2/**-25.2** | 105.0/**62.3** | 77.8/**49.7** | 0.40/**0.32** |
| | | | | | | |
| | Winter | GSv3.0 | 2.5/**10.0** | 41.2/**30.8** | 28.2/**24.0** | 0.73/**0.72** |
| | | post-processed | -0.5/**5.5** | 40.2/**29.8** | 25.8/**21.4** | 0.75/**0.75** |
| | December | GSv3.0 | 11.0/**12.4** | 29.5/**26.1** | 22.0/**21.1** | 0.67/**0.72** |
| Eurasia | | Post-processed | 0.0/**1.5** | 23.9/**18.1** | 14.5/**13.1** | 0.70/**0.76** |
| | February | GSv3.0 | 10.5/**15.5** | 36.5/**30.9** | 26.5/**24.5** | 0.75/**0.76** |
| | | Post-processed | 5.6/**11.1** | 35.5/**28.1** | 24.0/**21.1** | 0.74/**0.76** |
| | April | GSv3.0 | -30.2/**-16.8** | 59.6/**39.6** | 42.3/**30.4** | 0.72/**0.64** |
| | | Post-processed | -18.3/**-8.8** | 57.6/**41.4** | 39.8/**31.5** | 0.71/**0.63** |

**GENERAL COMMENTS**

Please consider the number of significant digits for the error statistics reported in the text, figures, and tables. What is the measurement precision of the snow density observations? Are we really confident in density to the tenth or hundredth of a kg/m3? Are we really confident in SWE to the hundredth mm?

Number of significant digits will be reduced.

A histogram of snow density values (development and validation data) would be useful context, especially for interpreting Figure 5.

Figure below will be added to the article as figure 2.

[Figure]

**Figure 2**. Histogram of implementation and validation densities for Eurasia for 2000-2009.

Figures 8-10 are all referenced in the text before Figure 7 (see Lines 289-304). Consider reorganizing the text or renumbering/reordering the figures.

Figures are reordered and figures 8-10 combined into one figure.

The Discussion section and the Conclusion section are mislabeled and should be Sections 4 and 5, respectively.

Thanks, this will be fixed.

There is some discussion-like comments mixed into the results section (e.g., Lines 293-294, 302-304). Please consider reorganizing.

These comments are moved to discussion section and chapter 3.1 is reorganized significantly (lines 321-358).

The RMSE was 43.5 mm and 42.5 mm for the baseline retrieval and retrieval post-processed with decadal densities, respectively. Figure 8 shows scatter plots for the whole northern hemisphere for the baseline (a) and the post-processed datasets (b) and as expected, post-processing reduces overestimation of SWE

values between 10 and 100 mm. Figure 8 also shows density scatter plots for the baseline and the post-processed datasets for Eurasia ((c), (d)) and North America ((e), (f)) datasets. The post-processed Eurasian dataset shows similar improvements as the overall northern hemisphere dataset, overestimation of SWE values between 10 and 100 mm has been mitigated. For Eurasia, post-processing improves estimations every month when SWE values up to 500 mm are considered and the biggest improvement in RSME happens in December (over 5 mm). When SWE values only up to 150 mm are considered, we see similar big improvement in December, but in April baseline retrieval produces better results than the post-processed dataset.

North American datasets also shows some improvements but not as clearly as the Eurasian dataset. For North America SWE values are improved for December and April when SWE values up to 500 mm are considered. When SWE values up to 150 mm are considered, improvements are seen only in December. Figure 9, which shows the mean retrieval error and standard deviation of the error for baseline and post-processed retrievals for the northern hemisphere, Eurasia, and North America, agrees with these findings. Mean errors for North America are similar for SWE values below 100 mm for the baseline and post-processed retrievals but for larger values, the mean errors of the post-processed dataset are smaller than the errors of the baseline set. For Eurasia and northern hemisphere, the mean error is systematically smaller for the post-processed dataset than for the baseline dataset. Figure 10 shows SWE maps for February 6, 2011, for the baseline and post-processed retrievals. Figure 10 shows also difference between these two SWE maps (baseline minus post-processed). Maps show how SWE values are lower (darker colours) in the post-processed maps and the map of differences reveals that post-processing causes bigger changes in Eurasia than in North America.

**TECHNICAL CORRECTIONS**

Line 34: Should be 'data are also available' since 'data' are plural.

Line 40: Should be 'in deep snow'.

Line 44: Should be 'observations'.

Edits mentioned will be included, thanks.

Line 59: The Maurice and Harold (1981) citation is not included in the References section. Please add.

Citation will be added.

Line 62: Please rearrange parentheses such that it reads 'Sturm et al. (2010)'.

Line 91: Is a subsection required here? There is a 2.1.1.1 but no 2.1.1.2, so this subsection title may not be warranted.

Line 101: Should be 'were used'.

Line 120: This should be 'Figure 2'.

Line 130: Add 'The' before North American.

Line 132: Should be 'observations'.

Edits mentioned above will be included, thanks.

Lines 132-133: Can you state why only SNOTEL stations in Alaska and the northwestern USA were selected, and not SNOTEL stations farther south in the USA?

Explanation added to line 138: Data from southern states is not included as most of the snow in these areas is in mountains which are excluded from the retrieval.

Line 147: Please rearrange the parentheses here such that they are for the year only (two cases).

Line 186: Add 'the' before 'variogram'.

Line 208: Add 'the' before 'baseline'.

All the edits mentioned above will be included, thanks.

Line 224 and Table 1: The period of validation reported here does not correspond to what is described earlier in Lines 100-105. The annual data do not appear to span 2000-2009, and the decadal was previously described as 1999-2009.

These inconsistencies will be fixed, the correct period is 2000-2009.

Line 229: Should be 'decadal' rather than 'decal'.

Line 243-244: Add 'separately' after 'performed' to indicate that the validation was done for two different cases of SWE values.

Line 254: Should be 'decadal' rather than 'decal'.

Edits mentioned above will be included, thanks.

Line 282: Is this supposed to be 'SWE values up to 500 mm and 150 mm' rather than 'density values'?

Yes, it is supposed to be SWE.

Line 291: Replace 'improves' with 'reduces'.

Line 329: Should be 'SWE retrievals' at the end of this line.

Line 330: Replace 'effects' with 'errors'.

Edits mentioned above will be included, thanks.

**TABLE AND FIGURE COMMENTS**

Figures 5, 8, 9, 10: The vertical axis should read 'Estimated' rather than 'Estimates'.

Figure 5 caption: Should be 'decadal' rather than 'decal'.

Edits mentioned above will be included, thanks.

Figure 7 would be more effective if a third panel was included that showed the different in SWE estimates from GSv3.0 vs. post-processed. In the current form, I can only tell subtle differences between the two maps.

Figure 7: Note the title above each panel is overlapping text below (90 deg W). Please correct.

Updated figure 7 below

[Figure]

**Figure 7.** SWE maps for baseline GSv3.0 retrieval (top left), post-processed retrieval (top right) and difference between GSv3.0 and post-processed retrieval (post-processed subtracted from the baseline) for February 6, 2011. The post-processed SWE values are lower as overestimation of small SWE values is reduced.

**REFERENCES**

Kelly, R. E., Chang, A. T., Tsang, L. and Foster, J. L.: A prototype AMSR-E global snow area and snow depth algorithm, IEEE Trans. Geosci. Remote Sens., 41(2), 230–242, doi:10.1109/TGRS.2003.809118, 2003.

Tedesco, M. and Narvekar, P. S.: Assessment of the NASA AMSR-E SWE Product, IEEE J. Sel. Top. Appl. Earth Obs. Remote Sens., 3(1), 141-159, doi:10.1109/JSTARS.2010.2040462, 2010.

---

## Author Response (AR2)

**Author's response**

We thank the referee again for his/her valuable comments on the manuscript. We have changed the title of the article to "Impact of dynamic snow density on GlobSnow snow water equivalent retrieval accuracy" and fixed all the mentioned typos.